# SkinCon: A skin disease dataset densely annotated by domain experts for fine-grained model debugging and analysis

**Roxana Daneshjou**[1][*]   **Mert Yuksekgonul**[2][*]
**Zhuo Ran Cai**[1]   **Roberto Novoa**[1]   **James Zou**[3]
[1] Department of Dermatology, Stanford University
[2] Department of Computer Science, Stanford University
[3] Department of Biomedical Data Science, Stanford University
{roxanad,merty,zrcai,rnovoa,jamesz}@stanford.edu

## Abstract

For the deployment of artificial intelligence (AI) in high risk settings, such as healthcare, methods that provide interpretability/explainability or allow fine-grained error analysis are critical. Many recent methods for interpretability/explainability and fine-grained error analysis use concepts, which are meta-labels that are semantically meaningful to humans. However, there are only a few datasets that include concept-level meta-labels and most of these meta-labels are relevant for natural images that do not require domain expertise. Previous densely annotated datasets in medicine focused on meta-labels that are relevant to a single disease such as osteoarthritis or melanoma. In dermatology, skin disease is described using an established clinical lexicon that allows clinicians to describe physical exam findings to one another. To provide a medical dataset densely annotated by domain experts with annotations useful across multiple disease processes, we developed SkinCon: a skin disease dataset densely annotated by dermatologists. SkinCon includes 3230 images from the Fitzpatrick 17k skin disease dataset densely annotated with 48 clinical concepts, 22 of which have at least 50 images representing the concept. The concepts used were chosen by two dermatologists considering the clinical descriptor terms used to describe skin lesions. Examples include "plaque", "scale", and "erosion". These same concepts were also used to label 656 skin disease images from the Diverse Dermatology Images dataset, providing an additional external dataset with diverse skin tone representations. We review the potential applications for the SkinCon dataset, such as probing models, concept-based explanations, concept bottlenecks, error analysis, and slice discovery. Furthermore, we use SkinCon to demonstrate two of these use cases: debugging mistakes of an existing dermatology AI model with concepts and developing interpretable models with post-hoc concept bottleneck models.

## 1   Introduction

As we work towards deploying artificial intelligence (AI) for images in high risk settings such as healthcare, methods that provide interpretation or explanation for human operators and allow fine-grained error analysis are important. Previous methods developed to address these issues in image analysis have relied on the use of high-level human concepts, which are semantically meaningful and provide a way to further analyze images (Ghorbani et al., 2019). For example, for a dataset of images

---

[*]Equal Contribution

36th Conference on Neural Information Processing Systems (NeurIPS 2022) Track on Datasets and Benchmarks.

that contains cats and dogs, concept labels may include "whiskers" or "collar". These concepts can then be used to provide interpretability/explainability for a model or to better understand when a model makes an erroneous prediction.

An example of how concepts are applied is concept bottleneck models, which rely on labeled concepts within image data in order to predict the class label in an interpretable manner by first predicting concepts from image inputs and subsequently predicting class labels from those concepts (Koh et al., 2020; Yuksekgonul et al., 2022). Abid et al. (2022) developed an approach, conceptual counterfactual explanations, which used human-explainable concepts to explain why a model misclassified a particular instance.

While there has been work on the automatic generation of concepts (Ghorbani et al., 2019; Yeh et al., 2020), many of these methods rely on datasets with labeled concepts for development and testing. Because both delineating what concepts to use and labeling concepts within an image is laborious, these kind of datasets are not as common. Currently, healthcare datasets with meta-labels that could be used as concepts include: 1) the Osteoarthritis Institute Knee X-ray dataset (OAI) which has knee X-rays of patients at risk for osteoarthritis with over 4,000 patients and more than 36,000 observations but only a subset of which have 18 clinical concepts related to osteoarthritis (Nevitt et al., 2006) 2) PH2 which has 200 images of skin disease with lesion segmentation and 7 clinical attributes (Mendonça et al., 2013) 3) derm7pt which has 1011 images with 7 clinical criteria associated with melanoma (Kawahara et al., 2018). However, all of these datasets focus on concept descriptions related to identifying a single disease process (osteoarthritis for OAI and melanoma for PH2 and derm7pt).

Dermatology serves as an ideal use case for concept-based labeling because clinicians have an established lexicon for describing skin lesions based on their texture, shape, and color (Bolognia et al., 2017). Dermatologists use these standardized terms to describe lesions to one another when discussing physical exam findings (Bolognia et al., 2017). Previous datasets in dermatology have focused on descriptors specifically related to melanoma and have also lacked diversity in skin tones (Mendonça et al., 2013; Kawahara et al., 2018). This more narrow focus may limit the potential for developing generalizable interpretability/explainability methods and tools for fine-grained error analysis.

**Our contributions**   With the current limitations in mind, we present SkinCon: a skin disease dataset densely annotated by domain experts (dermatologists). SkinCon includes 3230 images from the Fitzpatrick 17k skin disease dataset densely annotated with 48 clinical concepts, 22 of which have at least 50 images representing the concept. These same concepts were also used to label 656 skin disease images from the Diverse Dermatology Images (DDI) dataset, providing an additional external dataset. When combined with Fitzpatrick17k, we have 25 concepts with more than 50 images, and 32 concepts with more than 30 images. SkinCon is the first medical dataset densely annotated by domain experts to provide annotations useful across multiple disease processes. To illustrate the type of applications enabled by SkinCon, we show that we can use medical concepts learned from SkinCon to explain why AI models misdiagnose patients and also to develop more interpretable AI models.

## 2   Dataset

### 2.1   Data Sources

SkinCon is developed based on images from two existing datasets: Fitzpatrick 17k (Groh et al., 2021) and Diverse Dermatology Images (DDI) (Daneshjou et al., 2022). Both of these datasets are available for scientific, non-commercial use. Images from Fitzpatrick 17k originate from two publicly available online atlases, as described previously by (Groh et al., 2021). Skin tone breakdown for the subset of images included from Fitzpatrick 17k are in Table 1; skin tone breakdown for DDI was previously reported in Daneshjou et al. (2022) with of 208 images of FST (Fitzpatrick Skin Tone) I-II, 241 images of FST III-IV, and 207 images of FST V-VI.

Because Fitzpatrick 17k was scraped from online atlases, the level of dataset noise is higher (for example, Fitzpatrick 17k has non-skin images) than DDI, which had every image curated by a dermatologist. For Fitzpatrick 17k, we selected a random sampling of images; non-skin images or images without a clear lesion were filtered out. All images from DDI were used. Disease annotations

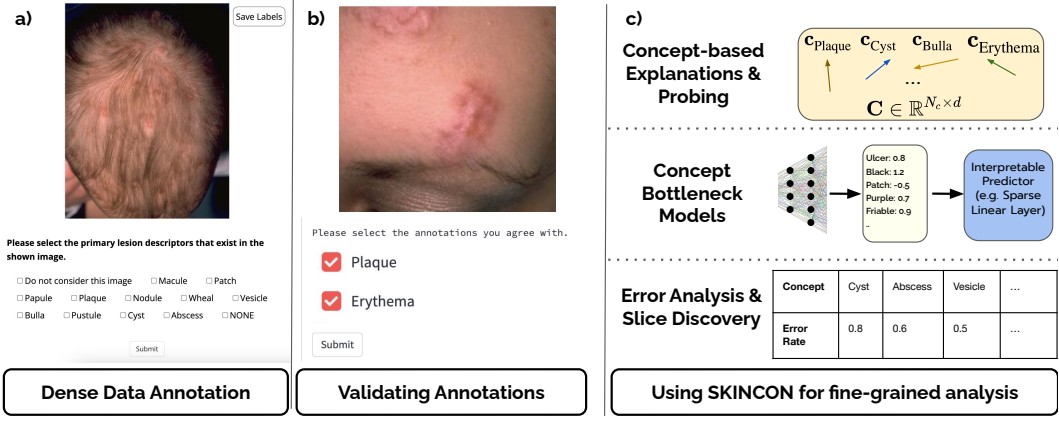

Figure 1: **An overview of the SkinCon dataset. a) Data labeling interface.** We provide the labeler with potential skin descriptors, and the labeler marks the ones that exist in the image. We ask labelers to consider 48 concepts, the list can be found in the Appendix. **b) Data validation interface.** We provide the validator with the currently labeled concepts, and we let them mark the ones they agree with. **c) Potential applications.** Using SkinCon, users can provide concept-based explanations, turn their models into post-hoc concept bottlenecks, or analyze the mistakes made by their models.

(fine grained diagnosis as well as benign versus malignant) and Fitzpatrick skin tone annotations were previously provided with each dataset. Fitzpatrick 17k disease annotations are not verified by skin biopsy; a description of the verification process can be found in (Groh et al., 2021). Fitzpatrick 17k skin tone was annotated by non-dermatologist human annotators using Scale AI using the 6-point Fitzpatrick skin tone scale. DDI diseases were all verified by skin biopsy; skin tone was labeled using three bins (Fitzpatrick I-II, III-IV, and V-VI) based on information from the clinical visit, demographic images, and lesion images and further validated by two dermatologists as described previously (Daneshjou et al., 2022).

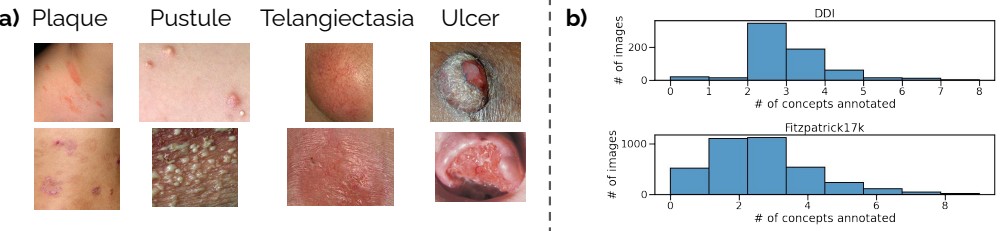

Figure 2: **Examples from the SkinCon dataset. a) Example concepts and samples.** We show two examples each for four prevalent concepts: Plaque, Pustule, Telangiectasia and Ulcer. **b) Statistics.** Here we provide the distribution of number of concepts per each image for the Fitzpatrick17k subset.

## 2.2 Data Concepts and Labeling

A unique aspect of dermatology is the existence of a lexicon used for describing skin disease findings; dermatologists undergo extensive training using these clinical terms which are used for both written and verbal communication to describe the appearance of skin lesions (Bolognia et al., 2017). These descriptors are meant to allow dermatologists to visualize the appearance of the disease process. For example, atopic dermatitis lesions might be described as "erythematous plaques with scale" while a skin cancer might appear as an "ulcerated erythematous nodule". These terms share information about the size, texture, shape, and color of the lesion. For instance, a plaque is a raised lesion that is greater than 1 cm in diameter but less than 1 cm in height; while a nodule is 1 cm or greater in height (Bolognia et al., 2017). Prior to reviewing the image data, two dermatologists with 5 and 6 years of clinical dermatology experience created a list of concepts based on the clinical lexicon used by dermatologists to describe skin lesions and with consultation of the terms listed in one

Table 1: Distribution of images over skin tones. The Fitzpatrick skin tone scale has been used by dermatologists and AI practitioners for assessing skin color. Fitzpatrick I-II represents white skin, III-IV represents olive and light brown skin, while Fitzpatrick V-VI represents dark brown and black skin.

| Fitzpatrick Skin Tone | #Images (Fitzpatrick17k) | # Images (DDI) |
|---|---|---|
| I-II | 1738 | 208 |
| III-IV | 1350 | 241 |
| V-VI | 467 | 207 |
| Unknown | 135 | - |

of the most widely used dermatology textbooks - Dermatology by Bolognia et al. (2017). The dermatologists were tasked with picking concept terms that would describe the most commonly encountered lesion shapes/size, color, and texture. Each dermatologist selected terms to add to the list, with both approving all terms prior to labeling. A total of 48 clinical concepts were selected from existing clinical terms. During the labeling process, these were grouped by primary lesion characteristics (which have to do with morphology) (Figure 1a), secondary lesion characteristics (which cover textural changes), additional shape information, and color. Then, each image was labeled with the present concepts using a standardized labeling interface by a single dermatologist with 6 years of dermatology experience (Figure 1). Since dermatologists are trained in these clinical descriptors, detailed explanations of each concept was not required. Note that images could have multiple concepts. Figure 2a shows a sampling of clinical concepts along with images that these concepts appear in. A label of "Do not use" was allowed to remove any images that were poor quality.

## 2.3 Validation

As a first validation step, a board-certified dermatologist validated 323 (10%) of the images (selected randomly); the validator agreed with $1056/1082 = 97.6\%$ of the concept annotations from the Fitzpatrick17k subset.

To get further validation, all of the images from the DDI dataset (656 images) and a random selection of 300 images from the Fitzpatrick dataset were independently labeled using the same labeling interface as was used in the initial labeling procedure, for a total of 956 images. 94% of these images were of sufficient quality across all labelers. Validation labels were provided by two dermatologists with 12 and 5 years of dermatology experience. Overall, we found that independent validators' annotations agreed with SkinCon labels 94% of the time – 92% for Fitzpatrick and 94% for DDI.

## 2.4 Statistics

Overall, we release 3230 clinical images from Fitzpatrick17k and 656 images from DDI datasets with meta-labels. We have 48 concepts, listed in the Appendix along with the number of images from each concept. 22 of these concepts have more than 50 images in Fitzpatrick 17k subset, and when combined with DDI, we have 28 concepts with more than 50 images. A summary of the distribution of number of concepts seen in an image in Fitzpatrick 17k subset can be found in Figure 2b; the x-axis bins represent the number of concepts annotated, while the y-axis shows the number of images that are annotated with that particular number of concepts. The distribution of images across Fitzpatrick skin tones can be found in Table 1.

## 3 Experiments with SkinCon

For all of the experiments, we use DeepDerm Daneshjou et al. (2022) model, which is trained on the data used by (Esteva et al., 2017) and based on Inceptionv3 (Szegedy et al., 2016). DeepDerm is trained on clinical images to predict whether a skin lesion is benign or malignant. Because there is data leak between the images used to train DeepDerm and the Fitzpatrick 17k dataset, we conduct all our tests on the completely independent DDI dataset. The baseline performance of DeepDerm on DDI is similar to what was previously reported in Daneshjou et al. (2022) (Table 2).

## 3.1 Task 1: Explaining model mistakes with SkinCon

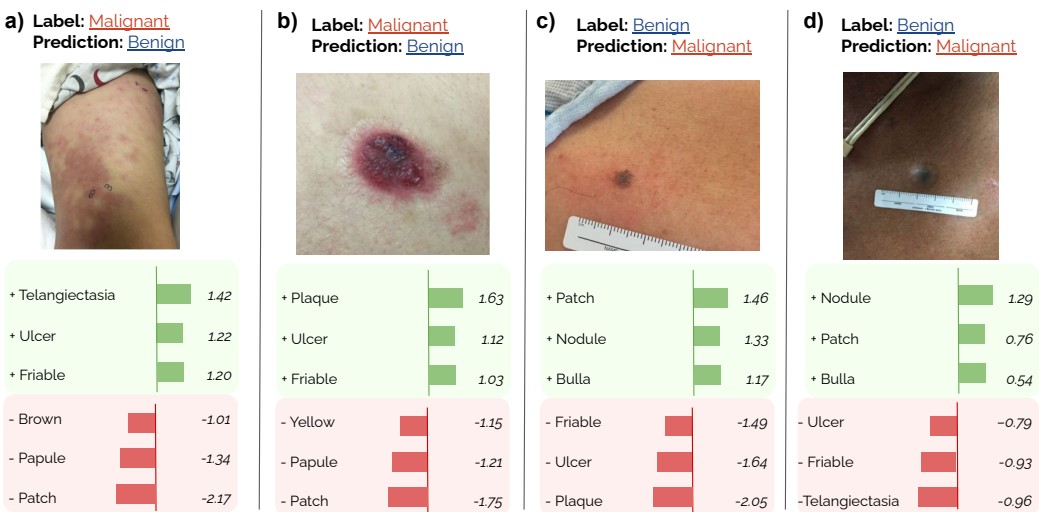

Figure 3: **Explaining model mistakes using SkinCon concepts.** Here we report concepts we found that would explain the model mistakes using Conceptual Counterfactual Explanations (CCE). Particularly, a large positive weight would mean adding the concept to the image would help fix the model mistake. Similarly, a large negative weight would mean removing the concept from the image would help fix the mistake.

To explain DeepDerm's mistakes, we follow Conceptual Counterfactual Explanations (CCE) (Abid et al., 2022). Particularly, CCE has two steps, as described below.

**Learning concepts:** We use Concept Activation Vectors(CAV) (Kim et al., 2018) to learn concepts for these models. Let $f : \mathcal{X} \to \mathbb{R}^d$ denote the layers up to the final layer of the model, which takes the image as an input and produces an embedding. For a given concept indexed by $i$, we collect two sets of embeddings $P_i = \{f(x_{p_1}), ..., f(x_{p_{N_p}})\}$, that contains the concept, and similarly negative examples $N_i = \{f(x_{n_1}), ..., f(x_{p_{N_n}})\}$ that do not contain the concept. We then train a linear SVM using $P_i$ and $N_i$ to learn the corresponding CAV. In our experiments, we use concepts with at least 50 images in the Fitzpatrick 17k subset, which amounts to using 22 concepts. For training the linear probes, we use 50 pairs of positive and negative images. In Appendix, we give an ablation where we explore the effect of number of samples used on the performance of the linear probes.

**Generating counterfactual explanations:** Using concept activation vectors, we use the CCE algorithm to generate counterfactual explanations. Briefly, CCE aims to add/remove concepts using intermediate activations to reason about model behavior. Given the embedding of a mistake, CCE finds a counterfactual example by modifying the embedding through adding a weighted sum of the learned concept activation vectors such that the model assigns high likelihood to the opposite label. Intuitively, this is done to understand what concepts should change in the sample in order for the model to change its prediction.

We refer the reader to Abid et al. (2022) for implementation details. A set of examples can be found in Figure 3. Particularly, a large positive weight would mean adding the concept to the image would help fix the model mistake. Similarly, a large negative weight would mean removing the concept from the image would help fix the mistake. For example, 3a shows that if this lesion would have had telangiectasias, ulceration, or friability, it would have been more likely to be properly classified as malignant. Overlying telangiectasias are often seen in skin cancers such as basal cell carcinoma. Friability is tissue that looks like it may easily bleed, with existing tears; this is a characteristic of abnormal skin in skin cancer. Ulcers are breakdown of the skin, which can be seen in skin cancer since there is abnormal tissue present. Thus, these concepts are all clinical features seen in malignancy (Bolognia et al., 2017).

Table 2: **Post-hoc Concept Bottleneck Models with SkinCon.** We report the model AUCs over the DDI dataset for both the original model and the PCBM. We observe that PCBMs with SkinCon is at least as good as, or in some cases better than the original model. We trained the PCBM 5 times with random seeds and report the standard errors next to the mean accuracies over 5 runs.

| Test AUC | DDI | DDI(I-II) | DDI(III-IV) | DDI(V-VI) |
|---|---|---|---|---|
| Original Model (DeepDerm) | 0.595 | 0.649 | 0.632 | 0.528 |
| PCBM with SkinCon | $0.639 \pm 0.001$ | $0.643 \pm 0.002$ | $0.727 \pm 0.002$ | $0.542 \pm 0.002$ |

Table 3: Importance of concepts for the concept bottleneck model. A large positive weight means larger contribution towards predicting an image as malignant. Similarly, a large negative weight means larger contribution towards predicting as benign.

| Concept Name | Concept Weight |
|---|---|
| Ulcer | 1.190 |
| Telangiectasia | 1.022 |
| Black | 0.746 |
| Purple | 0.489 |
| Friable | 0.307 |

| Concept Name | Concept Weight |
|---|---|
| Patch | -0.960 |
| Pustule | -0.836 |
| Scar | -0.711 |
| Dome-shaped | -0.130 |

### 3.2 Task 2: Interpretable models with concept bottlenecks

Can we implement inherently interpretable models using concepts? Following Post-hoc Concept Bottleneck Models (PCBM) (Yuksekgonul et al., 2022), we project all embeddings to a concept bottleneck. Specifically, the bottleneck denoted by $f_c : \mathcal{X} \to \mathbb{R}^{N_c}$ maps an input to an $N_c$ dimensional vector, where each dimension corresponds to a concept and $N_c$ is the number of concepts. Later, an interpretable predictor, such as a linear model is used to make the prediction. Particularly, for a sample $x$, the prediction would be $\boldsymbol{w}^T f_c(x) + b$. We can later analyze $\boldsymbol{w}$ to understand the importance of each concept per the model.

In this experiment, we use the 22 concepts used in Section 3.1(concepts with at least 50 images) to implement the bottleneck. The linear predictor of the PCBM is trained and evaluated on the Fitzpatrick17k dataset images that were not used to label concepts, where we split the remaining images into the training (2787 images, 80%) and test sets (697 images, 20%). Then, we test the performance on both a held-out set of Fitzpatrick 17k, and also the Disparities in Dermatology Dataset (Daneshjou et al., 2022). In Table 2, we report the overall results.

We observe that using 22 concepts in the bottleneck, we can recover the original model performance or better using PCBMs. While we trained on images across all skin tones to learn the concepts, we did analyze how PCBM performed across skin tone subsets on the DDI dataset. As seen in Table 2, for Fitzpatrick III-IV images, PCBM had an AUC of 0.727 compared to the original model AUC of 0.632, and in Fitzpatrick V-VI images, PCBM had an AUC of 0.542 compared to an AUC of 0.528 for the original model. On the interpretability side, we can investigate which of these concepts are important per the model. In Table 3, we observe the concepts that predict malignancy according to the model. Particularly, skin lesions described using the concepts *Ulcer*, *Telangiectasia*, *Black* are more likely to be predicted as a malignancy. Clinically, this makes sense, as black is a color seen in melanoma, cancerous skin is more likely to ulcerate, and telangiectasias are prominent in several skin cancers, most commonly basal cell carcinomas (Bolognia et al., 2017). However, concepts do not always align with clinical expectation. Here, *Scar* is a concept that is predictive of a benign lesion, and while scarring is often seen in non-malignant process, recurrent skin cancers appear near a scar from prior removal. This is further discussed in the Limitations section.

## 4 Potential Applications

There is a large variety of applications that require fine-grained annotations. In Table 4 we provide a non-exhaustive list of example applications that can use SkinCon.

| Application | Related Works |
|---|---|
| Probing models | (Alain & Bengio, 2016; Adi et al., 2016) |
| Concept-based Explanations | tCAV(Kim et al., 2018), CCE(Abid et al., 2022) |
| Concept Bottlenecks | CBM(Koh et al., 2020), PCBM(Yuksekgonul et al., 2022) |
| Error Analysis, Slice Discovery | (Eyuboglu et al., 2022; Chung et al., 2019) |

Table 4: Proposed applications for our dataset.

**Application-1: Probing models** The idea of training classifiers in internal representations of artificial/biological neural networks is prevalent across disciplines (Cox & Savoy, 2003; Alain & Bengio, 2016; Ivanova et al., 2021). In probing, the idea is to "probe" internal representations of models to predict target features by training classifiers. If a classifier can predict a target feature well, we can suggest that the particular feature is encoded in the representation space. It is important to note that having a high-performing probe does not imply that the target feature is later used by the model (for this purpose, see concept-based explanations below). Using our dataset, we can probe skin classifiers to understand which of these important concepts are encoded by the model.

**Application-2: Concept-based explanations** try to explain model behavior using human-interpretable concepts. Kim et al. (2018) aims to find out the sensitivity of model predictions to concepts in a bank, Akula et al. (2020); Abid et al. (2022) generates counterfactual statements to explain model behavior/mistakes. Different from probing, these methods aim to explain if a given concept is being used / important for model behavior. In all of these use cases, users need to pre-define a concept bank to probe the model behavior. Our work will allow dermatology models to be probed for a rich set of clinical descriptors. Similar in spirit, Lucieri et al. (2020) uses derm7pt Kawahara et al. (2018) and PH2 Mendonça et al. (2013) datasets to obtain concept banks to analyze skin lesion classifiers with CAVs(Concept Activation Vectors). In these datasets, they have a handful of concepts ($< 10$) labeled for a limited number of images ($1,011$ for derm7pt, $200$ for PH2). We showed in Section 3.1 that we can leverage SkinCon to meaningfully reason about model mistakes over the DDI dataset.

**Application-3: Concept Bottlenecks** projects the inputs onto a set of interpretable concepts, and later uses the concepts to make predictions. Concept Bottleneck Models (CBMs) (Koh et al., 2020) were first implemented using densely annotated training datasets and required concept annotations at training time. Later, PCBMs (Yuksekgonul et al., 2022) implemented this as a post-hoc procedure, where any neural network can be turned into a concept bottleneck model, given a concept bank. Our work will allow concept bottlenecks to be implemented in various dermatology use cases. We demonstrated in Section 3.2 that using SkinCon, we can recover and in some cases exceed model performance by converting a black-box model to a concept bottleneck. Furthermore, we can look at concept importance in the interpretable predictor to understand what concepts the model is relying on.

**Application-4: Error Analysis, Slice Discovery** aims to find coherent sets of mistakes to debug machine learning models. A model may have high overall accuracy but make systematic errors on particular subsets of the data, referred to as slices (Eyuboglu et al., 2022). Such differential performance has been noted in multiple applications of medical AI, which can have dangerous consequences for patients (Badgeley et al., 2019). For example, images of pneumothorax chest X-rays without chest drains (a treatment for pneumothorax) were more likely to be misclassified as a false negative (Oakden-Rayner et al., 2020). Fine-grained analysis methods such as slice discovery aim to identify these subsets or slices with differential performance. Concepts can serve as "ground truth" slices of data for testing slice discovery methods (Eyuboglu et al., 2022).

## 5 Related Works

### 5.1 Densely annotated general datasets

Densely annotated datasets that provide meta-labels that could be used as concepts are key for the development of interpretable/explainable methods and fine-grain analysis methods. There is a large set of general purpose datasets that are densely annotated. Much of the non-medical datasets in this realm do not require special domain expertise; however, there is also not an established lexicon for

describing these kinds of images, leaving the meta-labels to the discretion of the dataset creators. For example, the Caltech-UCSD Birds-200-2011 (CUB) dataset has 11,788 bird photographs representing 200 species and 312 binary attributes based on the appearance of the bird; sentences describing the birds have also been collected using Amazon Mechanical Turk to provide natural language descriptions (Wah et al., 2011). The Animals with Attributes (Xian et al., 2018) dataset provides 50 animals with 85 attributes, and is used in testing few-shot/zero-shot classification. Similar in spirit, Visual Genome (Krishna et al., 2017) is a large-scale crowdsourced dataset of 108k images densely-annotated with object/attribute/action information. Metashift (Liang & Zou, 2022) was derived from Visual Genome with the purpose of testing for distribution shifts, introducing context-based shifts in various classes. (Bau et al., 2017) proposes Broden Visual Concepts dataset, which combines several densely annotated datasets to derive 63,305 images with 1197 visual concepts, where concepts are scenes, objects, texture, color, material or parts.

## 5.2 Densely annotated medical datasets

One of the largest medical datasets in this space is the the Osteoarthritis Institute Knee X-ray dataset (OAI) which has knee X-rays of patients at risk for osteoarthritis with over 4,000 patients and more than 36,000 observations (Nevitt et al., 2006). However denser annotations are only provided for those images that meet a particular threshold of osteoarthritis severity, as these concepts are based on clinical findings seen in osteoarthritis, such as subchondral sclerosis and joint space narrowing (Koh et al., 2020). PH2 has 200 images and derm7pt has 1011 images of skin lesions; both have annotations for 7 clinical attributes associated with a diagnosis of melanoma (Mendonça et al., 2013; Kawahara et al., 2018). Because of the need for domain expertise in annotations, medical datasets generally focus on concepts related to a single diagnosis rather than broader concepts that could be applied in a more generalized manner. Recently, explainable diagnostic methods were developed using DermXDB, which has 554 dermatology images annotated by experts (Jalaboi et al., 2022).

# 6 Conclusions

We release our dataset at `https://SkinCon-dataset.github.io`. We developed SkinCon to meet the need for datasets that will facilitate the creation of methods for interpretability/explainability and fine-grained error analysis for high risk settings such as healthcare. Unlike previous densely labeled datasets in medicine, SkinCon does not use concepts based on identifying a single diagnosis but rather relies on the general lexicon of terms used by dermatologists for describing a wide range of skin diseases.

## 6.1 Contributions

Our main contribution is the SkinCon dataset, which provides dense, domain expert annotations on two sets of images drawn from Fitzpatrick 17k and DDI. This is the first densely annotated medical dataset with concepts that can be broadly applied to describe multiple disease states rather than focusing on the features of a single diagnosis. To this end, we include 48 concepts, which represents the highest number of meta-labels for any medical dataset. We demonstrate how this dataset can be used for two important concept-based use cases: explaining model mistakes (Section 3.1) and posthoc concept bottleneck models (Section 3.2). Additionally, we review potential methodological use cases for datasets with dense concept labels such as SkinCon (Section 4).

## 6.2 Limitations and Future Work

While we pull from two distinct data sources that cover a range of skin diseases, these are not comprehensive of all dermatological disease. For example, we find that *Scar* is a concept that predicts a benign lesion (Table 3). While scars are seen in many benign diseases, recurrent skin cancers appear near a scar from prior removal and certain types of skin cancer such as morpheaform basal cell carcinoma and dermatofibrosarcoma protuberans can have a scarlike appearance. This dataset likely did not include a large number of skin cancers that were due to recurrence or rarer scar-like presentations of skin cancers. Healthcare datasets are often imbalanced due to real world differences in disease prevalence; as a result, SkinCon is also imbalanced in its concepts. For example, papules and plaques are commonly seen; while nodules, which are skin lesions that are > 1 cm in height are

seen less commonly. Additionally, information about patient sex and gender, race, and age were not included with either Fitzpatrick 17k or DDI. This limits our ability to do fine-grained error analysis across protected classes. For future work, we hope to expand the size and scope of this dataset by densely labeling additional skin disease images. As we observe in DDI and Fitzpatrick 17k annotations, each dataset and task can have a different distribution of concepts and diseases. Real-world practitioners should be aware of such differences, and fine-grained annotations facilitate validations of models and datasets in different settings. SkinCon provides the foundation for this future work.

### 6.3 Impact on society

AI datasets created for dermatology have traditionally lacked diverse skin tones (Daneshjou et al., 2021, 2022), which is a significant concern since algorithms developed from datasets without diverse skin tones perform worse on diverse skin tones (Daneshjou et al., 2021). For example, both the PH2 and derm7pt datasets lack dark skin tones (Mendonça et al., 2013; Kawahara et al., 2018). Here, we pulled from datasets that specifically included diverse skin tones, though even still, Fitzpatrick V-VI represents a smaller portion of the data than I-IV (Table 1). Moreover, we used the Fitzpatrick skin tone scale since the data was previously labeled with this scale; however, this scale may not have enough granularity to capture the full range of human skin diversity (Okoji et al., 2021). Alternative scales have been suggested but none have been validated for AI skin tone labeling tasks.

The concept labels here are based on previously defined clinical terms that were pulled from Dermatology by Bolognia et al and reviewed by 2 dermatologists (Bolognia et al., 2017). However, this does not mean that these terms are inclusive of all possible descriptor terms. While terms have agreed upon definitions – such as a macule is a flat lesion less than 1 cm and a patch is a flat lesion greater than 1 cm – the assessment of size in an image is subjective and therefore prone to some noise.

The images used come from prior datasets, which each report their terms of use and collection practices. We did not have an IRB for this study since we were labeling publicly available data. We note that since these are clinical images of disease processes on human skin, there may be sensitive or distressing images.

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
