# OpenReview forum: "SkinCon: A skin disease dataset densely annotated by domain experts for fine-grained debugging and analysis"
_NeurIPS.cc/2022/Track/Datasets_and_Benchmarks — NeurIPS 2022 Datasets and Benchmarks _

### Official Review · Reviewer_yQAD · 2022-07-20
**Interesting dataset with too little information about the labelling protocol and the levels of noise**

**Rating:** 6
**Confidence:** 5
**Clarity:** The paper is well written, with a few…

**Strengths:**

- The dataset contains interesting, densely annotated data that will increase the explainability and debugging capabilities of skin cancer diagnosis models.
- The images in the dataset cover a variety of skin tones.
- A list of use-cases for this dataset is demonstrated and prototyped, acting as guidelines for future research.


**Weaknesses:**

- The methodology offers little insight into the levels of noise present in the data:
    - No intra-rater analysis has been performed, thus making it difficult to ascertain the level of noise present in this dataset.
    - While the authors have performed a very limited validation of the label correctness, the review process cannot be interpreted as inter-rater agreement. A correct inter-rater agreement analysis would require the raters to be blinded to each other’s evaluations in order to avoid bias. The current validation setup is likely due to overstate the level of agreement between the two raters.
- More information about the labelling protocol should be provided to allow for reproducing this work and to better understand the labels:
    - No information is given about the level of education and expertise of the original labeller and of the reviewer.
    - How were the 48 clinical concepts chosen?
    - The caption in Figure 1 states that the labellers were asked to consider 48 concepts, but Figure 1a) shows only 11 concepts. Could you explain this discrepancy?
    - Figure 1a) shows that labellers had an option to discard images by selecting “Do not consider this image”. How often did this happen? Why were images being discarded?
- In the abstract, the authors claim that this is the first medical dataset annotated by domain experts. However, in Section 5.2 they present several other medical datasets with additional or dense annotations, including in dermatology. Moreover, two similarly densely annotated dermatological datasets are not included in the related work: please refer to [DermX](https://openreview.net/forum?id=wBOBL5b-aa9&noteId=EeXtzTjYNBi) and [DermIS](https://www.dermis.net/dermisroot/en/home/index.htm) for more details.
- The explanation mechanism suggested in Section 3.1 and the interpretability method presented in Section 3.2 require that the models perform reasonably well at detecting the clinical concepts present in the image. Could you report this performance to prove that this dataset can indeed be used in the proposed manner? Without a high detection performance, the explanations and the interpretations may not be valid from a domain perspective.
- No information is given about how the validation set was selected.
- While the Fitzpatrick skin tone distribution is discussed, more demographics metadata (e.g. age, sex) is needed to understand whether the dataset is balanced over the demographics.


**Additional Feedback:**

- For ease of reading, I would recommend moving Table 1 (or an abridged version of Table 1) to the main body of the paper.
- The caption for Figure 1b) may be slightly confusing: I suggest changing “we let them mark the ones they disagree with” to “we let them mark the ones they agree with”, so that the caption matches the phrasing depicted in the image.
- Figure 2b) shows the distribution only for the Fitzpatrick17k subset. Could you add a similar figure for the DDI data?
- Several abbreviations are used without having been introduced a priori:
    - L77: FST
    - L160: pCBM
    - L195: tCAV
    - L199: CBMs
- Typos:
    - L11-12: the double usage of “to provide” is redundant
    - L128: redundant “of the”, and an uncapitalized “deepderm”.
    - L211: capitalize the x in “x-rays”
    - L246: the link is missing a /
    - L259: "moddeels"
    - L283: “their terms of use anc collection”
- Could you mention how many images was the PCBM linear predictor trained on (L160-161)?
- Could you mention the exact number of concepts that were labelled in tCAV (L195)?


**Correctness:**

The dataset construction and the use-cases seem mostly correct, although more information about the labelling protocol is needed.

**Documentation:**

- Not enough information about data collection protocol is given to allow for reproducing this work.
- No information is given about availability and maintenance plans for the data.


**Ethics:**

More information about the age and the sex distribution in this dataset should be provided.

**Relation To Prior Work:**

While the authors describe the differences between their work and densely annotated datasets from other medical fields, two densely annotated dermatological datasets with similar annotations are not mentioned or discussed.

**Summary And Contributions:**

This paper introduces SKINCON, a densely annotated dermatological dataset which includes annotations of 48 clinical concepts performed by one dermatologist for 3886 images. Another dermatologist reviewed less than 10% of these annotations, with high levels of agreement between the two raters. The authors demonstrate SKINCON’s value by using it on two tasks: model mistake explanation and model interpretability. Four more possible applications are proposed.

---

> ### Author Response · Authors · 2022-08-20
> **Author Response to Reviewer yQAD - Part 1**
>
> Dear Reviewer yQAD,
>
> We would like to thank you for your responsible and thoughtful review, and your thorough questions on the annotations. Below are our detailed comments regarding your questions:
>
> **Annotations / Validation**:
> Thank you very much for this important comment. Based on this feedback, we have recruited two independent validators to assess a subset of the data. Please see the updates in the Validation section:
> “As a first validation step, a board-certified dermatologist validated 323 (10%) of the images, where the validator agreed with 1056/1082 = 97.6% of the concept annotations from the Fitzpatrick17k subset. To get further validation, all of the images from the DDI dataset 656 images and 300 random images from the Fitzpatrick dataset were independently labeled using the same labeling interface as was used in the initial labeling procedure, for a total of 956 images. 94% of these images were of sufficient quality across all labelers. Validation labels were independently provided by two dermatologists with 12 and 5 years of dermatology experience. *Overall, we found that independent validators' annotations agreed with SKINCON labels 94% of the time -- 92% for Fitzpatrick and 94% for DDI.*”
>
> **Labeling Protocol**:
> Thank you for this valuable feedback. We have updated that the original labeler had 6 years of dermatology experience. We have updated to clarify that the original concepts were chosen by 2 dermatologists with 5 and 6 years of dermatology experience in consultation with Dermatology by Bolognia et al, one of the most commonly used Dermatology textbooks. We have updated that during this process, they selected the most common terms used for describing lesion shape/size, texture, and color with both approving the final list. Regarding Figure 1a, we have clarified that this panel does not show all the labels since labels were grouped to make labeling easier, “During the labeling process, these were grouped by primary lesion characteristics (which have to do with morphology) (Figure 1a), secondary lesion characteristics (which cover textural changes), additional shape information, and color. We have clarified in the text that “Do not consider this image” was an option during labeling to remove images that were of insufficient quality for labeling. We have also clarified in the limitations section that we do not claim that this is inclusive of every possible clinical descriptor term but is meant to start as a foundation.
>
>
> **Descriptions of prior datasets**:
> Thank you for your comments. In the abstract, we are highlighting that we are the first medical dataset whose annotations are not focused on a single disease (such as features that distinguish osteoarthritis or melanoma); however, given that the wording may cause confusion, we have removed it. We have added DermX in our previous works section and cited the preprint; however, we found that there was no working public access to the dataset for DermX: https://github.com/ralucaj/dermx. For DermIS, we found a website (https://www.dermis.net/dermisroot/en/home/index.htm), but could not locate dense annotations or a paper describing the annotations. If the reviewer has seen a paper for DermIS, we would deeply appreciate the link so we can cite it.

---

> > ### Author Response · Authors · 2022-08-20
> > **Author Response to Reviewer yQAD - Part 2**
> >
> > **Experimental Details**: We added detailed results in the Appendix for how the number of samples and the validation set are picked for learning concepts. Briefly, we use leave-10-out cross-validation to decide on the number of samples to use for concepts, where we see that performance saturates around 50 images. Increasing further would mean using a very small number of concepts, and decreasing would mean having less performant probes. Please see Appendix Page 1 Section A.1 for further details.
> >
> > **Typos / Abbreviations / Figure Updates**: We thank you for your careful reading, and we fixed all of the suggested typos/lack of definitions of the abbreviations /details. We further added dataset statistics separately to Figure 2 (Page 3) and Appendix Table 2 (Appendix Page 3).
> >
> > **While the Fitzpatrick skin tone distribution is discussed, more demographic metadata (e.g. age, sex) is needed to understand whether the dataset is balanced over the demographics.**: Thank you for this important point. Neither Fitzpatrick 17k nor DDI published demographic metadata with the public release of the datasets. For DDI, it was reported that Fitzpatrick I-II and V-VI images were matched by age and gender. We are using previously publicly released images for this paper.
> >
> > Thank you again for your feedback and suggestions, which have improved the paper. If our detailed response and revision have answered your questions, we would greatly appreciate it if you would consider increasing your score. Please let us know if you have further questions and we are happy to follow up!

---

> > > ### Comment · Reviewer_yQAD · 2022-08-29
> > > **Assessment changed to 6**
> > >
> > > Thank you for the thorough reply to my concerns, I will change my assessment to a 6. In the future, I an inter-rater and intra-rater study might be relevant, as well as adding demographics metadata to the dataset.

---

### Official Review · Reviewer_TJGe · 2022-07-22
**A densely annotated skin image dataset to achieve model interpretability**

**Rating:** 8
**Confidence:** 5
**Clarity:** The presentation of the paper is good.

**Strengths:**

The interest in adopting dermatology AI models has increased in the last few years. Unfortunately, most of the developed models are black box models, where understanding the fail cases and solving them is very hard. SKINCON dataset can help  users/developers/clinicians in understanding the behaviour of these black box models as well as improve their performance.

For efficient training of the AI models using the concept labels, the authors used methods developed for another applications by the co-authors. This shows that the authors have full control over the data.

The images in the dataset represent different skin colors in a balanced manner.

Beyond the dataset, the provided applications to test the dataset are well choosen


**Weaknesses:**

It is not clear how are the concepts decided. The authors mentioned that the concepts are based on clinical descriptor terms used to describe skin lesions. However, do the authors think, the opinion of two dermatologists is enough for consensus.

The labeling is done by a reader and then finetuned (corrected) by an expert dermatologist. Again, do the authors think labeling by two readers is reliable enough?

How do the authors ensure that the proposed concepts are generalizable to a larger cohort of dermatologists?

Concept labels are not available neither at the github site or the supplementary zip file (or at least they were not easily available so the reviewer missed them)

**Additional Feedback:**

NA

**Correctness:**

Besides the clarification needs mentioned in the weakness section, the dataset is constructed in a sound way
The presented benchmark and evaluation methods sound correct too

**Documentation:**

The authors provide all the necessary information in the supplementary. They also provide a GitHub link for code and data. During the time of the review, there was a problem with the DDI dataset repository but the reviewer believes that this will be fixed soon.

The reviewer could not find the concept labels of the images, neither in the github site or the supplementary zip file

**Ethics:**

Authors used public datasets, therefore ethics documents special for this dataset is not needed.

**Relation To Prior Work:**

Yes

**Summary And Contributions:**

Authors present dense annotations for two different skin image datasets, aiming to increase model interpretability of AI models. The dataset is composed of skin images taken from the well-known Fitzpatrick 17K dataset (3230 images) and Diverse Dermatology Images (DDI) dataset (656 images). The photos are densely annotated for 48 different (predefined) concepts (an image may take on more than one concept) defined by two dermatologists.
The authors also illustrate potential applications that can utilize such dense annotations. They used the dataset to understand better the mistakes of an existing dermatology image classification model and interpret the same classification model via post-hoc concept bottlenecks.

---

> ### Author Response · Authors · 2022-08-20
> **Author Response to Reviewer TJGe**
>
> Dear Reviewer TJGe,
>
> We would like to thank you for your responsible and thoughtful review. Below are our detailed comments regarding your questions:
>
> **Concept labels**: You can find these on our website in the dataset section. Please see the lines containing e.g. [Download SKINCON Fitzpatrick17k annotations by clicking here. ].
>
> **It is not clear how are the concepts decided. The authors mentioned that the concepts are based on clinical descriptor terms used to describe skin lesions. However, do the authors think, the opinion of two dermatologists is enough for consensus.**
> Thank you for the opportunity to clarify. As mentioned in Section 2.2, these clinical descriptors are widely taught and are available in the most widely used dermatology textbook, Dermatology by Jean Bolognia. We have also updated our limitations, “The concept labels here are based on previously defined clinical terms that were pulled from Dermatology by Bolognia et al and reviewed by 2 dermatologists. However, this does not mean that these terms are inclusive of all possible descriptor terms.”
>
> **The labeling is done by a reader and then finetuned (corrected) by an expert dermatologist. Again, do the authors think labeling by two readers is reliable enough?**
> Thank you for this comment. We have updated our validation section with additional work:
> ““As a first validation step, a board-certified dermatologist validated 323 (10%) of the images, where the validator agreed with 1056/1082 = 97.6% of the concept annotations from the Fitzpatrick17k subset. To get further validation, all of the images from the DDI dataset 656 images and 300 images from the Fitzpatrick dataset were independently labeled using the same labeling interface as was used in the initial labeling procedure, in a total of 956 images. 94% of these images were of sufficient quality across all labelers. Validation labels were provided by two dermatologists with 12 and 5 years of dermatology experience. *Overall, we found that independent validators' annotations agreed with SKINCON labels 94% of the time -- 92% for Fitzpatrick and 94% for DDI.*”
>
> **How do the authors ensure that the proposed concepts are generalizable to a larger cohort of dermatologists?**: Thank you for this comment. We have clarified in the text that these clinical concepts are derived from already established clinical terms, which are covered in several dermatology textbooks. We used one of the most popular textbooks, Dermatology by Bolognia et al during the development of the proposed concepts.
>
> We thank you again for your insightful comments, and please let us know if you have any further questions.

---

### Official Review · Reviewer_WEWC · 2022-07-24
**Dense (exhaustive) 48-feature clinical dermatology annotations in two image datasets, important contribution to the literature**

**Rating:** 7
**Confidence:** 4

**Strengths:**

- the authors clearly demonstrate the benefit of collecting and using "within-image" (meta/feature) labels to further examine and inspect AI model mis-classification cases
- exhaustively labeling this number of images for 48 potential features must have been an incredibly time consuming effort, and I commend the authors for this undertaking!
- providing the dermatology community with insights into how and why models may make mistakes is of critical importance to reduce misdiagnoses, and the authors' work makes in important contribution in that direction


**Weaknesses:**

Most importantly, the annotations are incredibly skewed and the distribution fairly dissimilar across the two datasets, warranting some caution as to how anything "learned" from these specific labels translates to a population of images used in a different setting/context

Minor points:
- section 2.4/tables: it would be great if the authors could provide a table of the 48 labels with the frequency (absolute or fraction) of images in which the label was annotated, to give readers a sense of the density (and likely reliability) of each of the 48 labels; this would also reveal a discrepancy between the two datasets (e.g., papule is labeled in 36.2% of images in one, but 64.5% of images in the second dataset), putting into question the influence of baserates (distributional properties) during training
- the paper is missing an evaluation of how clinicians would utilize the information presented by such methods -- e.g., given an AI prediction class label and the additional information of, say, a PCBM output, how does this affect clinician reliance on/trust in the AI predicted label?
- table 2 suggests a whopping improvement in performance specifically for FST III and IV; I would like to see some explanation as to why that might be the case (possibly the concepts with the densest annotations occur most frequently in those skin types?); this might also lead to a limitation of the presented dataset as primarily benefiting AI aided diagnoses in certain populations (FST III and IV)?


**Additional Feedback:**

- lines 110-113: does that mean that the core concept annotations (for the whole dataset) are from a single dermatologist expert? while the validation (of 10% of images) reached a very high agreement (>95%), I wonder to what extent this might also be due to the validating expert having received similar/identical training to the original annotating expert, and how a group of experts might reveal higher (or similar) concordance?
- there are a few typos throughout the manuscript (in particular, section 5.2 contains several superfluous words)


**Clarity:**

Overall, the paper is well written, with some minor points here:

- line 63: why was the threshold of 50 chosen (as opposed to, say, 20; at which 13 additional concepts would have qualified)?
- lines 140-143: while I appreciate the difficulty summarizing the referenced work (Abid et al., 2021), I believe it would be beneficial for the reader to appreciate how the SVM output (of samples on which the classifier was not trained) is used to derive the weights; if, for instance, a sample image's embeddings (latent variables) produce a projection into one of the two clusters (P for positive and N for negative training samples), with a distance relating to c (SVM parameter for separability of the hyperplane), how does this value then relate to the weight given a specific concept?
- as much as the "counterfactual hypothetical" (if one were to add a concept to an image, the label would be correct) is useful, it does ultimately not explain why the model *missed* the concept
- instead of phrasing as "adding the concept to the image would help fix the model mistake", I would suggest "would increase the likelihood of the model assigning the opposite label" (in practice, the ground truth is often not known, and it is important for clinicians to appreciate the aspect of a model "missing" a concept in an image, and that if the model had detected the concept, it *might* have assigned the opposite label)
- what then would be important to know is maybe some indicator of *how likely* (for each missed label) it would be that the model would alter the predicted label (does a weight of 1, for instance, mean a shift by 1 unit log-odds?)
- lines 152 following: while the term "post-hoc concept bottleneck models" appears (as a whole) in the caption of table 2, it is (in the full text) never linked directly to the PCBM acronym; it might be useful to do so in either line 160 or 154 -- by adding the word "model" to the sentence, and then adding (PCBM) in parentheses.


**Correctness:**

The overall dataset construction (two CSV files in the main) seems sound and easily usable across softwares (e.g. python/pandas, etc.).

Minor points (please correct):
- line 64 and table 1 describe 656 DDI images, whereas the downloaded CSV file (from https://skincon-dataset.github.io/index.html only contains 636 entries)
- line 65 describes 28 concepts with 50+ images; combining the two CSV files, my count is 25 concepts with 50+ images


**Documentation:**

The dataset itself is the annotations generated (binary indicator for each of the 48 potential labels being present in each of the images). No concerns.

**Ethics:**

No concerns.

**Relation To Prior Work:**

The article discusses relevant prior work. No concerns here.

**Summary And Contributions:**

- the paper and dataset represents the work of manually labeling 3,866 images (from two separate datasets) for 48 concepts (> 180k total feature annotations), together with potential applications of informing users of conventional (binary) classification as to whether or not specific features, if detected, would alter the classification (predicted label)
- this is an interesting and useful contribution to the literature that helps readers become aware of the kinds of mistakes (dermatological) AI models can make, and how to inquire into those mistakes

---

> ### Author Response · Authors · 2022-08-20
> **Author Response to Reviewer WEWC - Part 1**
>
> Dear Reviewer WEWC,
>
> We would like to thank you for your responsible and thoughtful review. We especially appreciate you taking the time to download & look at our annotations, and giving thoughtful insights on the details of the dataset and presentation of the paper. Below are our detailed comments regarding your questions:
>
> **Most importantly, the annotations are incredibly skewed and the distribution fairly dissimilar across the two datasets, warranting some caution as to how anything "learned" from these specific labels translates to a population of images used in a different setting/context**
> Thank you for the comments - certain lesion morphologies (which were encoded as concepts) are always going to be more common - such as papules because they describe many different kinds of disease processes. However, despite there being differences in concept distributions between the training and test sets, we were able to use the concepts learned from the training data to create clinically reasonable counterfactual examples and post-hoc concept bottleneck models that had similar performance to the base model in the test data. But concept labels are not only useful for “learning” concepts; they are also useful for fine-grained error analysis. For example, a skin diagnosis model may consistently mislabel images containing rare concepts, and this may be useful for understanding why a diagnosis was missed.
>
> **Finer-grained statistics on the distribution of concepts**: Thanks for pointing this out, we have updated Table in the Appendix (see Page 3).
>
> **Number of annotations**: 20/656 images are marked as “do not use image”, either because the lesion is unclear, or it is not possible to label concepts in the image. In the new version, we also include this column to ensure that this is reflected properly. Further, you are correct that we have 25 concepts with >= 50 images, and we have updated our manuscript to reflect this. Thank you very much for going through the effort of downloading and checking the dataset!
>
> **The choice of 50 as the threshold**: In Appendix Page 1 Section A.1, we add an ablation where we vary this threshold and explore the performance of the linear probes. Shortly, the performance of the linear probe is lower for around 30 samples, and performance saturates around 50 samples. Further increasing the number of samples would also crucially limit the number of concepts we can use with the analysis. We hope this answers your question, please see Section A.1 in the appendix for further details.
>
> **the paper is missing an evaluation of how clinicians would utilize the information presented by such methods -- e.g., given an AI prediction class label and the additional information of, say, a PCBM output, how does this affect clinician reliance on/trust in the AI predicted label?**
> We agree with the reviewer that understanding how clinicians would interact with AI and explainable AI is an area of interest. The field of clinician-AI interaction is still in its infancy with recent papers like Tschandl et al Nature Medicine 2020 and Vodrahalli et al AIES 2022 focusing on interactions with black box models. The evaluation of how concepts affect clinician reliance on and trust in the AI-predicted label requires a large-scale prospective trial with many clinicians, which is beyond the scope of this paper. However, this dataset is exactly what could enable such a trial in the future, particularly in the realm of dermatology.
>
> **as much as the "counterfactual hypothetical" (if one were to add a concept to an image, the label would be correct) is useful, it does ultimately not explain why the model missed the concept**
> Thank you for this comment – we agree that ultimately explaining why the model missed a concept is important and something the field is working towards. We have clarified with this sentence, “Intuitively, this is done to understand what concepts should change in the sample in order for the model to change its prediction.”

---

> > ### Author Response · Authors · 2022-08-20
> > **Author Response to Reviewer WEWC - Part 2**
> >
> > **lines 110-113: does that mean that the core concept annotations (for the whole dataset) are from a single dermatologist expert?**
> > Thank you for this comment. We updated the validation section with two independent validators.  Additionally, all three trained at different dermatology programs.
> >
> > “As a first validation step, a board-certified dermatologist validated 323 (10%) of the images, where the validator agreed with 1056/1082 = 97.6% of the concept annotations from the Fitzpatrick17k subset. To get further validation, all of the images from the DDI dataset 656 images and 300 images from the Fitzpatrick dataset were independently labeled using the same labeling interface as was used in the initial labeling procedure, for a total of 956 images. 94% of these images were of sufficient quality across all labelers. Validation labels were provided by two dermatologists with 12 and 5 years of dermatology experience. **Overall, we found that independent validators' annotations agreed with SKINCON labels 94% of the time -- 92% for Fitzpatrick and 94% for DDI.**”
> >
> > We thank you again for your insightful comments, and please let us know if you have any further questions.

---

> > ### Comment · Reviewer_WEWC · 2022-08-25
> > **Would appreciate another note in the limitations (6.2) section as a "future direction"**
> >
> > I very much appreciate the response and changes to the manuscript, and believe that this pretty much addresses everything.
> >
> > One small point I would like the authors to consider is that the *difference* in skew (of concepts/features across the two datasets used) suggests that concepts collected on additional data (a hypothetical third dataset) might yet lead to differences, and as such would likely also lead to different weights for concepts on diagnoses, suggesting that ultimate robustness of this method will emerge over time. This is not so much a limitation per se, but simply makes it necessary to take this initial (SKINCON) dataset as a starting point towards developing/collecting additional, multi-faceted annotations (where the tasks and source material vary sufficiently to prevent the specifics that went into the two datasets in question to affect the ultimate purpose of "un-biasing" AI decision making). I would appreciate if this was mentioned as a "future direction" (either in the limitations/6.2 section or elsewhere).
> >
> > Otherwise, congratulations again on your work!

---

> > > ### Author Response · Authors · 2022-08-26
> > > **Author Response to WEWC**
> > >
> > > Thank you so much for this insightful comment. We have updated our limitations and future directions section in response to this comment, "For future work, we hope to expand the size and scope of this dataset by densely labeling additional skin disease images. As we observe in DDI and Fitzpatrick annotations, each dataset and task can have a different distribution of concepts and diseases. Real-world practitioners should be aware of such differences, and finer-grained annotations facilitate validations of models and datasets in different settings. SKINCON provides the foundation for this future work."

---

### Official Review · Reviewer_McEh · 2022-07-25
**proposed dataset consists of dermoscopic images with skin diseases, densely annotated by domain experts for developing interpretability/explainability methods and fine-grained error analysis.**

**Rating:** 7
**Confidence:** 4
**Correctness:** The claims described in the paper app…
**Clarity:** Yes

**Strengths:**

1. The paper is well-written, and the authors presented a clear conceptualisation associated with the paper.
2. Provided a fine clinical annotation, which is lacking in previous data settings for two existing datasets, Fitzpatrick 17k and Diverse Dermatology Images (DDI)
3. Considering the factors for skin lesions based on their texture, shape, and colour, which doctors use for determining the type of skin lesion is used and is  an important factor in medical image analysis
4. Datasets incorporated the concept of skin tone with the samples to reduce the effect of data bias


**Weaknesses:**

1. Data imbalance is present across the clinical labels, which can affect the model efficacy
2. The number of dermatologists used to annotate is missing in the paper. Also, how the annotations were balanced among them? Any coefficient such as Cohen's Kappa etc. is, used?
3. How can we adhere to the ‘concept-based’ decision and say it's responsible for the particular decision? For instance, predicting a lesion as benign or malignant if it's not present in the sample, as all the mentioned  ‘concepts’ can be present in a single sample, particularly in applications 1, 2 and 3. (section 4).


**Additional Feedback:**

Writing mistakes:
Line 136: I think positive for Pi is missing.


**Documentation:**

Yes.

**Ethics:**

Not required. The authors used the existing publicly available dataset.

**Relation To Prior Work:**

Yes.

**Summary And Contributions:**

The authors proposed dataset consists of dermoscopic images with skin diseases, densely annotated by domain experts for developing interpretability/explainability methods and fine-grained error analysis. The images are taken from two prior publicly available skin lesion datasets, Fitzpatrick 17k and Diverse Dermatology Images (DDI).

---

> ### Author Response · Authors · 2022-08-20
> **Author Response to Reviewer McEh**
>
> Dear Reviewer McEh,
>
> We would like to thank you for your responsible and thoughtful review. Below are our detailed comments regarding your questions:
>
> **Data imbalance is present across the clinical labels, which can affect the model efficacy:** Healthcare by its nature is an imbalanced task, and in particular, among skin lesions which come in a variety of different shapes. For example, melanoma is far less common than basal cell and squamous cell skin cancers. Skin lesion shapes such as papule and plaque are more commonly seen; while larger lesions such as nodules are less common. Additionally, the balance of the labels in any given population is decided based on population characteristics. While this can affect model efficacy, it reflects real-world issues, where rare diseases have fewer examples. We have added this to our limitations, “Healthcare datasets are often imbalanced due to real-world differences in disease prevalence; as a result, SkinCON is also imbalanced in its concepts. For example, papules and plaques are commonly seen; while nodules, which are skin lesions that are > 1 cm in height are seen less commonly.”
>
> Further, note that this is precisely why one may need fine-grained annotations. With the help of SKINCON, we can explicitly test images for their behavior under this imbalance, with the help of skin tone/concept annotations.
>
> **The number of dermatologists used to annotate is missing in the paper. Also, how the annotations were balanced among them? Any coefficient such as Cohen's Kappa etc. is, used?**
> Thank you very much for these comments. We have updated that the original annotations were done by one dermatologist with 6 years of clinical dermatology experience. During the review period, we performed additional validation by having a second label produced **independently** for 300 images from the Fitzpatrick annotations (~10%) and all of the DDI annotations [656]. This validation was done by two dermatologists (5 and 12 years of experience). Across the 956 validated images, we have 94% agreement in annotations between the independent validator and the labeler.
>
> **How can we adhere to the ‘concept-based’ decision and say it's responsible for the particular decision? For instance, predicting a lesion as benign or malignant if it's not present in the sample, as all the mentioned ‘concepts’ can be present in a single sample, particularly in applications 1, 2, and 3. (section 4).**
> Thank you very much for these thoughtful comments. Concepts allow semantic explainability and opportunity for probing further. For example, with counterfactual explanations, we can see if the addition or subtraction of a concept would push a particular image across the decision boundary. This does not suggest that the concept is *solely* responsible for a benign or malignant decision but allows us to see what concepts by their presence or absence can affect the output of the model, using *the counterfactual*. Concept bottleneck models allow us to make predictions with similar performance to black-box models, but with the added benefit of having global concept weights that quantify the importance of a given concept.
>
> We thank you again for your insightful comments, and please let us know if you have any further questions.

---

### Official Review · Reviewer_8yKc · 2022-07-26
**Innovative meta-label benchmark for dermatology with limited evaluation**

**Rating:** 7
**Confidence:** 5

**Strengths:**

- Innovative dataset for meta-label analysis in dermatology
- Thorough analysis and discussion of applications
- Paper is well-written and easy to understand


**Weaknesses:**

- Table 1: what is the skin tone for III-IV (see lines 94-107)

- A single annotator (a board-certified dermatologist) labelled images from two existing datasets. In medical data annotation, it is often the standard to have multiple annotators. It is also not clear how much experience this dermatologist has.

- Analysis os conducted using a single model (DeepDerm with InceptionV3 backbone).

- Only a single concept analysis method (Concept Activation Vectors (CAV)) is studied.

- Lack of visual explanations. There is no discussion of methods such as CAM or Grad-CAM, and not analysis of concept heat maps. In Figure 9 of CCE (Abid 2021), it seems like it is possible to generate heat maps and it is unclear why the authors do not discuss/show them.

- Limited analysis of embeddings. In Section 3.2, embeddings are analyzed using a linear predictor. Given that the number of concepts is small, visuals such as colored cluster maps would be helpful to understand whether embeddings are disentangled or not

- In Section 3.2, analysis of concepts by skin tone is discussed. However, it is difficult to understand the conclusions. It would help to include quantitative results the describe concept weight change or other statistics by the Fitzpatrick skin tone and/or number of images.


**Additional Feedback:**

The motivation and contributions of the paper are excellent. As described in weaknesses, the main issue is the lack of discussion to visual explanation methods and limited evaluation.

**Clarity:**

Paper is well-written and easy to understand.


**Correctness:**

The contribution of the paper is a mainly a dataset. Given the limited number of methods studied, it is unclear if any conclusions can be drawn.

In Line 115: “we release 3230 clinical images”. This is an overstatement. These clinical images are available in a public datasets and the paper provides additional labels to them.


**Documentation:**

Yes, documentation is very thorough. Dataset is released via Github. It would be helpful to add links to permanent data storage such as OSF ( https://osf.io) and add an information how to seek support/answer questions. Additional information (code, data-sheet, and data statistics) are provided in the supplementary material.  Intended uses are well-described.

**Ethics:**

No ethical concerns.


**Relation To Prior Work:**

Paper describes prior work in densely annotated general and medical datasets fairly well. Certain concept explanation works are also described, although a dedicated analysis of meta-concept analysis is missing. It would be good to give a broad overview. Also, the following reference is missing:

Jose Oramas, Kaili Wang, Tinne Tuytelaars. Visual Explanation by Interpretation: Improving Visual Feedback Capabilities of Deep Neural Networks. ICLR 2019


**Summary And Contributions:**

The paper introduces SKINCON, a set of expert annotations on images in existing dermatology datasets (Fitzpatrick 17k (Groh 2021) and Diverse Dermatology Images (DDI)) (Daneshjou 2022)). Motivated by concept analysis work that aim to analyze whether a model encodes a meta-label, i.e., an annotation attached to data, but for which the model is not trained for, the dataset allows analysis of explanation of model mistakes. As one of the first meta-learning applications in medical imaging and dermatology, the dataset would be extremely valuable both to the CV/ML and the medical image community.

---

> ### Author Response · Authors · 2022-08-20
> **Author Response to Reviewer 8yKc**
>
> Dear Reviewer 8yKc,
>
> We would like to thank you for your responsible and thoughtful review and for raising insightful questions about the annotations / downstream effects. Below are our detailed comments regarding your questions:
>
>
> **Table 1: what is the skin tone for III-IV (see lines 94-107):** Thank you so much for pointing this out. We have modified it to read, “Fitzpatrick I-II represents white skin, III-IV represents olive and light brown skin, while Fitzpatrick V-VI represents dark brown and black skin.”
>
> **“A single annotator (a board-certified dermatologist) labeled images from two existing datasets. In medical data annotation, it is often the standard to have multiple annotators. It is also not clear how much experience this dermatologist has.”**
> Thank you for this comment. Depending on the particular label task, the number of annotators may differ. For example, the Fitzpatrick 17k dataset had additional validation of only 3% of their dataset by additional review (Groh et al, CVPR 2021).  In this case, as we are focusing on lesion morphologies and not diagnoses, label noise may be more permissive. A single dermatologist with 6 years of dermatology experience labeled the initial dataset. In response to the review, we collected an independent validation of a random 10% of the Fitzpatrick annotations (n=300 images) and the entirety of the DDI dataset [656 images]. The validation labels were provided by two additional dermatologists with 12 and 5 years of dermatology experience. Overall the annotations are consistent across the dermatologists. We have updated the manuscript with these clarifications and additions.
>
> **In Section 3.2, analysis of concepts by skin tone is discussed. However, it is difficult to understand the conclusions. It would help to include quantitative results the describe concept weight change or other statistics by the Fitzpatrick skin tone and/or number of images.**
> Thank you for the opportunity to clarify. We have added a clarification in the manuscript, “While we trained on images across all skin tones to learn the concepts, we did analyze how PCBM performed across skin tone subsets on the DDI dataset. As seen in Table 2, for Fitzpatrick III-IV images, PCBM had an AUC of 0.727 compared to the original model AUC of 0.632, and in Fitzpatrick V-VI images, PCBM had an AUC of 0.542 compared to an AUC of 0.528 for the original model.”
>
> **Analysis with a single model / visual explanations**
> Unfortunately, in the dermatology space, there are few examples of model backbones being shared with publication (Daneshjou et al, JAMA Dermatology 2021). Furthermore, Abid et al. found that saliency maps of skin lesion classifiers (e.g. using GradCAM) did not capture medically salient features. All of these are lesion images, and it is unclear what is to be gained by just using saliency maps, i.e. these will likely highlight lesions as salient parts of the image, and would not help us understand whether the model is paying attention to skin descriptors / color / pattern. Abid et al. demonstrated that it is challenging to understand these features with a saliency map.
>
>
> We thank you again for your insightful comments, and please let us know if you have any further questions.

---

> > ### Comment · Reviewer_8yKc · 2022-08-28
> > **Thank You**
> >
> > Thanks for your response. The additional study and the analysis of agreement is very convincing and useful. I have updated my rating.
> >
> > I wonder whether it is a surprise that the agreement between annotator and independent validator is so high (94%)? Is is that the nature of this task that makes such a a high agreement possible?

---

### Meta-Review · Area_Chair_FTDL · 2022-09-09

**Recommendation:** Accept
**Confidence:** 3

**Metareview:**

This paper provides an augmented dataset with additional annotations, intended to enhance interpretability and meta analysis of images for skin disease classification.

Pros:
i) The work is well motivated, justified and executed

ii) The dense annotation will significantly help test ML models against this dataset.

Cons:
i) Variability of annotations themselves can determine generalizability of the learned models and subsequent interpretations

ii) Currently it appears that few models have been test for analysis on the dataset.

Nonetheless there seems to be a consensus among reviewers that this dataset is indeed valuable. Hence I am recommending an accept.

---

### Decision · Program_Chairs · 2022-09-16

Accept